# Geo-Refine: Geometry–Appearance Synergy for Robust Single-Image 3D Scene Generation

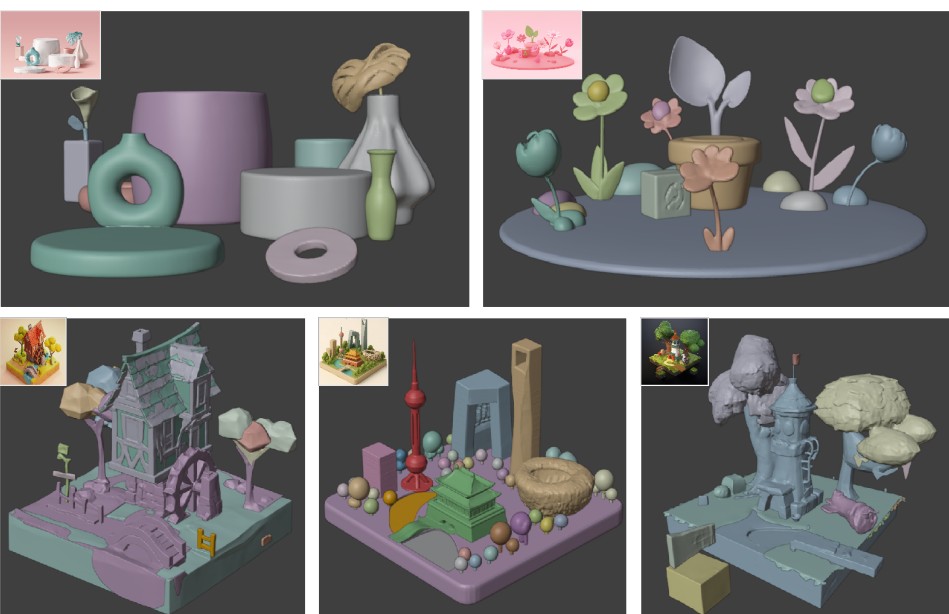

Figure 1: Geo-Refine generates a complete 3D scene from a single RGB image by coupling geometry–appearance preprocessing and appearance consistency—with a two-stage voxel–mesh localization that first reasons about global layout and then refines mesh poses for collision-free, physically plausible multi-object arrangements.

## Abstract

We introduce Geo-Refine, a single-image 3D scene generator that couples geometry–appearance preprocessing with a two-stage voxel–mesh localization pipeline to produce physically valid, visually complete multi-object scenes. Unlike prior methods that either overfit to image priors or rely on sequential post-hoc segmentation, Geo-Refine follows a unified, end-to-end formulation. Conditioned on one RGB image, it first extracts clean object regions through high-precision masking, directional color-spill suppression, and multi-view appearance consistency, then jointly optimizes object placement and fine mesh alignment. The global layout is cast as an energy-guided voxel reasoning problem that enforces projection evidence, ground support, and semantic co-location, while a subsequent mesh-level refinement stage guarantees collision-free, contact-accurate geometry. Experiments on diverse indoor and outdoor benchmarks show consistent gains in CLIP, VQ, and GPT-4 metrics, along with sharper geometry, stable object interactions, and improved multi-view fidelity over state-of-the-art image-to-3D baselines. These results highlight the value of Geo-refine for reliable single-image 3D scene synthesis and understanding.

# 1 INTRODUCTION

3D scene generation has emerged as a central problem in computer vision and graphics, with growing impact on applications such as digital content creation, game development, and robotics, which aims to synthesize objects and scenes composed of multiple semantically meaningful meshs. Unlike fused object generation, layout modeling enables downstream editing, physical reasoning, and compositional manipulation, making it a critical capability for scalable 3D scene understanding and synthesis (Huang et al. (2025); Ye et al. (2025); Hu et al. (2024); Yu et al. (2024)).

Current 3D generative methods often produce holistic meshes without explicit structures (Li et al. (2025); Yang et al. (2024); Wu et al. (2024a)). This limitation stems from their design: most approaches are optimized for global geometry reconstruction, but lack explicit mechanisms for decomposing objects into interpretable components. As a result, generated meshes are difficult to edit, without physical plausibility in contact regions, and fail to support higher-level reasoning about object functionality.

Several recent works attempt to address this by introducing decomposition pipelines (Yang et al. (2025); Liu et al. (2025a); Lyu et al. (2024); Jiang et al. (2025); Li et al. (2024b)). They typically segment fused scenes into incomplete meshes or layouts and perform per-mesh reconstruction. However, they suffer from two limitations. First, reliance on external segmentation priors—e.g., 2D vision models or pretrained networks—propagates errors: failures in segmentation irreversibly degrade generation quality. Second, sequential per-mesh processing is inefficient, with inference cost scaling linearly with the number of meshes, limiting scalability to complex scenes. We propose rethinking the pipeline via end-to-end, mesh-based 3D generation.

Our framework synthesizes an arbitrary number of disjoint meshes in a fixed-time budget, exploiting the observation that while contacting regions create ambiguity, disjoint meshes can be generated in parallel. To this end, we introduce an **independently-mesh-packing** strategy that maximizes space utilization while preventing unintended fusions between contacting items. We further formulate mesh grouping as a bipartite contraction problem, enabling a **voxel-mesh hybrid localization** that maintains a fixed output length while remaining fully compatible with latent denoising generative models. Building on this representation, each generated mesh is subsequently assembled into a coherent full 3D scene, preserving the geometric fidelity of individual components while capturing global spatial arrangements.

Extensive experiments demonstrate that our framework not only achieves superior quality and efficiency compared to prior baselines, but also provides explicit meshed that facilitate fine-grained editing, enforce physical plausibility, and support flexible scene-level manipulations. We demonstrate both quantitative gains in CLIP/VQ/GPT-4 metrics and qualitative improvements in generating semantically meaningful, manipulable 3D meshs. Our main contributions are summarized as follows:

- We present **Geo-Refine**, a unified framework for **single-image 3D scene generation** that jointly models global layout and fine-grained geometry without any external 2D/3D segmentation priors.

- We develop a **geometry–appearance preprocessing module** that integrates high-precision object masking, directional color-spill suppression, and multi-view appearance consistency to provide clean, coherent object inputs.

- We introduce a **two-stage voxel–mesh localization scheme**: an energy-guided voxel reasoning stage for coarse global placement, followed by mesh-level refinement that ensures collision-free alignment and physically valid contact geometry.

- We demonstrate consistent improvements in semantic fidelity, structural coherence, and visual quality over state-of-the-art single-image 3D baselines across diverse indoor and outdoor benchmarks.

## 2 RELATED WORK

### 2.1 3D SCENE GENERATION

Research on 3D scene generation can be grouped into three complementary directions: **isolated object-level generation**, **holistic multi-object scene synthesis**, and **physical or relational reasoning**.

**Isolated Object-Level Generation.** Representative isolated object-Level generation methods (Jun & Nichol (2023); Liu et al. (2023); Shi et al. (2024); Liu et al. (2024); Pan et al. (2025); Shen et al. (2025)) achieve strong geometry and texture quality. However, they do not explicitly model inter-object relations or scene-level context, so extending them to multi-object scenes often results in inconsistent layouts, collisions, or implausible arrangements.

**Holistic Multi-Object Scene Generation.** These works aim to directly synthesize entire scenes while jointly reasoning about layout, geometry, and appearance. (Hu et al. (2024)) predicts image-conditioned layouts and instantiates 3D assets. (Yu et al. (2024)) adopts diffusion-based priors to generate semantically consistent indoor and outdoor layouts. (Feng et al. (2023)) exploits vision–language models to infer spatial arrangements. (Li et al. (2023)) extends generative placement to outdoor environments. These methods typically lack fine-grained part-level control and may produce unrealistic local interactions, especially for cluttered or complex scenes.

**Physical and Relational Reasoning.** To ensure physically plausible and structurally coherent scenes, recent physical and relational reasoning work integrates explicit reasoning or post-optimization. (Pan & Liu (2025)) introduces reinforcement-based rewards encoding human-like placement rules. (Chen et al. (2025))formulates layout refinement as a graph optimization problem Hybrid multi-view and depth constraints, as in (Zhang et al. (2024; 2025)), further enhance geometric consistency. These approaches often require computationally expensive optimization or multi-stage refinement and still cannot fully guarantee high-fidelity object geometry.

### 2.2 3D DENOISING GENERATIVE MODELS

3D-native denoising models for conditional 3D generation have seen substantial progress in recent years. Research efforts focused on uncompressed 3D representations, such as point clouds (Li et al. (2024a); Qu et al. (2023); Liu et al. (2025b); Kong & Wan (2025); Lan et al. (2025); Vogel et al. (2024)), volumetric grids (Rasoulzadeh et al. (2025); Pinheiro et al. (2024); Maillard et al. (2024); Wu et al. (2024b)), and Neural Radiance Fields (NeRFs) (Gu et al. (2023); Chen et al. (2023); Chan et al. (2023); Jun & Nichol (2023); Höllein et al. (2024)). These representations face limitations when applied to small or sparse datasets, often resulting in poor generalization and suboptimal quality. For instance, direct volumetric diffusion models struggle with scalability and resolution constraints, while NeRF-based diffusion methods are prone to view inconsistency and high computational overhead.

Architectural innovations further advance this paradigm. (Yang et al. (2024)) introduces high-resolution latent embeddings to enhance surface detail reconstruction, while (Wen et al. (2025)) employs recursive 3D-aware diffusion to improve consistency across iterative generations. (Hu et al. (2024); Yu et al. (2024)) demonstrate that latent denoising frameworks can scale to full-scene generation, incorporating layout priors and semantic conditioning. (Huang et al. (2025)) extends diffusion to multi-instance 3D generation, supporting compositional scene synthesis from single-view input. Meanwhile, survey works (Kong et al. (2025); Chen & Wang (2024)) emphasize the synergy between denoising generative models and efficient rendering backends such as 3D Gaussian Splatting (Chen et al. (2024); Ververas et al. (2024)), highlighting their importance for high-quality and editable synthesis.

In this work, we extend these 3D latent denoising models to support mesh-level generation and physically plausible scene composition. By combining compact latent encoding with multi-view consistency constraints and hybrid localization, our approach achieves superior fidelity in both geometry and appearance, while remaining computationally scalable.

## 3 METHODOLOGY

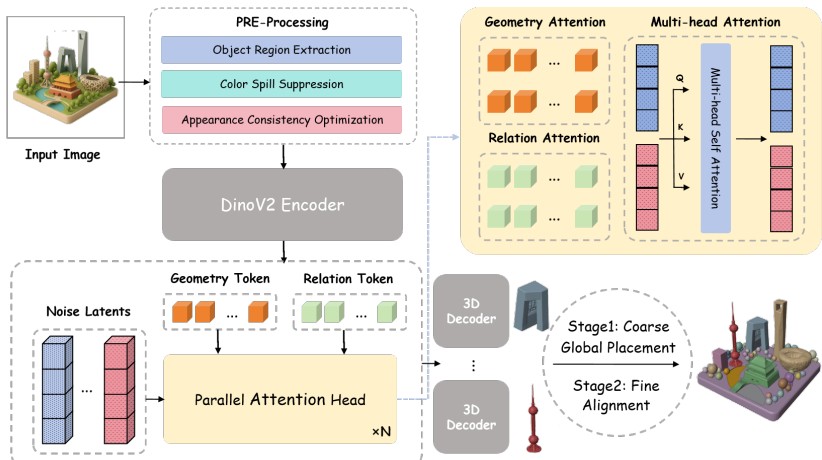

Figure 2: **Overall architecture of Geo-Refine.** Our model performs geometry–appearance preprocessing for clean object inputs, then encodes the image with DINOv2 features and applies parallel geometry–relation attention to capture local details and global layout. Fused tokens are decoded into high-fidelity 3D meshes with scene-level placement.

### 3.1 OVERALL STRUCTURE

As shown in Figure 2, our method generates a complete 3D scene from a single image by jointly modeling object geometry, global spatial relations, and cross-object contextual cues. Using DinoV2 encoder (Oquab et al. (2024)), a preprocessed input image $c$ is into a dual-stream latent representation: **Geometry tokens** $g_i \in \mathbb{R}^{K_g \times C}$ for fine object shape, **Relation tokens** $r \in \mathbb{R}^{K_r \times C}$ for holistic arrangement. The global latent set is defined as a combination:

$$Z = \{(g_i, r)\}_{i=1}^N \in \mathbb{R}^{N(K_g + K_r) \times C}. \tag{1}$$

**Parallel Attention** Each transformer block contains three parallel attention branches: Geometry Attention $A_{\text{geom},i}^h$, Relation Attention $A_{\text{rel}}^h$, and Multi-Head Context Attention $A_{\text{mh}}^h$. We concatenate the latent tokens as $[g_1; \ldots; g_N; r]$ and define the attentions as follows:

$$A_{\text{geom},i}^h = \text{softmax}\left(\frac{Q_i^h(K_i^h)^\top}{d_h}\right), \tag{2a}$$

$$A_{\text{rel}}^h = \text{softmax}\left(\frac{Q_{\text{rel}}^h(K_{\text{rel}}^h)^\top}{d_h}\right), \tag{2b}$$

$$A_{\text{mh}}^h = \text{softmax}\left(\frac{Q_{\text{mh}}^h(K_{\text{mh}}^h)^\top}{d_h}\right). \tag{2c}$$

Outputs from the three branches are summed with learnable weights and then passed through 3D Decoder to generate independent, integrated and physically plausible meshes. Cross-attention to encoded features $f(c)$ is injected into all three branches to maintain alignment with the conditioning image.

**Latent Flow Training** Given ground-truth latent $Z_0$, we follow similar designs from the rectified flow model (Liu et al. (2022)). The trained latent $Z_t$ is perturbed with Gaussian noise $\epsilon \sim \mathcal{N}(0, I)$ at time $t$:

$$Z_t = tZ_0 + (1 - t)\epsilon. \tag{3}$$

The transformer predicts a velocity field $v_\theta$, and the training loss is

$$\mathcal{L}_{\text{flow}} = \mathbb{E}_{Z_0, \epsilon, t} \|(\epsilon - Z_0) - v_\theta(Z_t, t, f(c))\|_2^2. \tag{4}$$

The decoded meshes $M_i$ and initial poses $\pi_i(0)$ initialize the subsequent coarse–fine optimization stages (Sec. 3.3), where $E_{\text{coarse}}(\{\pi_i\})$ and $E_{\text{mesh}}(\pi_i)$ provide collision-free placement and high-fidelity refinement.

By adding a parallel multi-head context branch to the geometry–relation dual attention, our network captures local, global, and cross-object dependencies simultaneously. This capability is crucial for robust image-to-3D scene generation and for producing accurate inputs to the downstream physics-aware placement pipeline.

## 3.2 Geometric and Appearance Preprocessing

While recent works in 3D scene synthesis (Zhang et al. (2024); Lyu et al. (2024); Ardelean et al. (2025)) have achieved impressive results, their performance is often bottle-necked by noisy or inconsistent object inputs. To address these challenges, we introduce a geometry and appearance preprocessing module composed of three innovations: high-precision object region extraction, directional color spill suppression, and multi-view appearance consistency optimization.

We formulate object masks as a prior-constrained optimization:

$$\hat{M}_j = \arg \min_{M_j} \ \mathcal{L}_{\text{seg}}(M_j, I) + \lambda \, \mathcal{L}_{\text{spatial}}(M_j, \mathcal{R}), \tag{5}$$

where $\mathcal{L}_{\text{seg}}$ ensures fidelity to the input image $I$ and $\mathcal{L}_{\text{spatial}}$ incorporates scene-level priors $\mathcal{R}$, such as object–ground contact or occlusion hierarchy. We further emphasize high-curvature regions and refine boundaries to preserve fine-grained edges.

Non-uniform color bleeding from lighting and reflections are severe problems with original images. To mitigate these, we model observed colors as:

$$C(x) = C_{\text{obj}}(x) + \alpha(x) \cdot S(x), \quad \alpha(x) = \sigma\big(\nabla I(x) \cdot d_{\text{light}}(x)\big), \tag{6}$$

where $\alpha(x)$ modulates spill contribution based on local gradients and illumination direction. This selectively attenuates unwanted hues while preserving intrinsic textures.

Object appearance is associated with feature discrepancies:

$$\mathcal{L}_{\text{cons}} = \sum_{(v_1, v_2)} \big\| \phi(F_{v_1}^i) - \phi(F_{v_2}^i) \big\|_2^2, \tag{7}$$

where $\phi(\cdot)$ encodes appearance and boundary features. Color normalization and temporal coherence regularization are applied to videos.

Together, these components produce clean and consistent object inputs for the voxel–mesh hybrid localization stage. Detailed procedures, including iterative refinement and sampling strategies, are provided in Appendix A.2.

## 3.3 Scene-level Voxel–Mesh Hybrid Localization

In order to place multiple objects into a coherent scene layout, we propose a two-stage voxel-mesh hybrid localization scheme. The design aims to (i) perform efficient global layout reasoning that avoids large-scale inter-object collisions and (ii) preserve local geometric fidelity and contact relationships through mesh-level refinement. It is inspired by advances in volumetric scene modeling and multi-object layout optimization (Chen et al. (2023); Zhao et al. (2024); Shi et al. (2024)).

**Preliminaries.** Let $\mathcal{O} = \{o_1, \ldots, o_N\}$ denote the set of $N$ object meshes extracted from the preprocessing stage (Sec. 3.2). Each object $o_i$ is associated with a high-resolution mesh $\mathcal{M}_i$ and an initial pose (translation $t_i$, rotation $R_i$, scale $s_i$). We denote $\mathcal{V}(\mathcal{M}, r)$ as a voxelization operator that maps mesh $\mathcal{M}$ to a binary occupancy grid at voxel resolution parameter $r$; the resulting voxel set for object $i$ is $V_i = \mathcal{V}(\mathcal{M}_i, r)$.

### 3.3.1 Stage I: Coarse global placement via voxel reasoning

We first compute a low-resolution voxel representation for each object:

$$V_i = \mathcal{V}(\mathcal{M}_i, r_{\text{coarse}}), \quad i = 1, \ldots, N, \tag{8}$$

Figure 3: **Scene-Level Voxel–Mesh Hybrid Localization Pipeline.** The method operates in two sequential stages. **Stage 1: Coarse Global Placement** (left) estimates a low-resolution voxel representation for each object and performs a global layout optimization $\Phi(\cdot)$ guided by placement energy $E_{\text{place}}$. **Stage 2: Fine Alignment** (right) illustrate how mesh-level details are progressively aligned with the scene to achieve accurate geometry and consistent physical interaction.

with $r_{\text{coarse}}$ chosen to trade off fidelity and efficiency. Using voxel placements we can efficiently evaluate overlaps between objects. For a pair $(i, j)$, denote the overlap volume (number of intersecting voxels) under candidate poses $\pi_i, \pi_j$ as

$$\text{Overlap}_{ij}(\pi_i, \pi_j) = \big| \pi_i(V_i) \cap \pi_j(V_j) \big|. \tag{9}$$

Define a collision graph $G = (V, E)$ where vertices correspond to objects and an edge $(i, j) \in E$ exists if $\text{Overlap}_{ij} > 0$ under the current poses. We associate an edge weight $w_{i,j}$ that quantifies the severity of the collision between objects $i$ and $j$.

The coarse placement problem is formulated as minimizing a global energy:

$$E_{\text{coarse}}(\{\pi_i\}) = \sum_i E_{\text{place}}(\pi_i) + \beta \sum_{(i,j) \in E} w_{ij} \Phi(\pi_i, \pi_j), \tag{10}$$

where:

- $E_{\text{place}}(\pi_i)$ encodes unary placement priors (e.g., keep object centroid near an initial layout estimate, respect floor contact or semantic anchors);

- $\Phi(\cdot)$ is a collision penalty (e.g., quadratic or robust penalty) that penalizes non-zero overlap.

**Unary Placement Term $E_{\text{place}}(\pi_i)$.** The first component of Eq. 10, $E_{\text{place}}(\pi_i)$ is associated with several complementary priors that jointly encourage each object $i$ to occupy a semantically and physically reasonable location in the reconstructed scene while remaining faithful to the input image. It enforces **projection consistency**, requiring that the 3D mesh under pose $\pi_i$ projects back to the image with a silhouette overlapping the detected 2D mask or bounding box, thereby tying the 3D reconstruction to observable evidence. **Ground-support constraint** further encourages the lowest surface to contact a valid supporting plane (such as the estimated floor or a detected table-top), prevents floating placements and promotes physical stability. In addition, **semantic-relation prior** leverages category-specific spatial statistics so the global arrangement reflects common regularities. Finally, **scale-and-orientation ingredient** penalizes implausible size changes or tilts by anchoring each object's dimensions and upright direction to distributions predicted by a single-view 3D estimator. These complementary cues ensure that $E_{\text{place}}$ encodes both image-level evidence and scene-level commonsense, driving the coarse optimization toward physically valid, semantically coherent placements before the finer mesh-level refinement of later stages.

We minimize Eq. 10 using a combination of greedy updates and small continuous pose adjustments: at each iteration, we (i) detect the highest-weight edge $(i^*, j^*)$, (ii) attempt a minimal translation of the object with lower placement cost to reduce overlap, and (iii) update the collision graph. This iterative procedure converges quickly in practice and yields a collision-free (or low-collision) global layout at voxel resolution.

### 3.3.2 STAGE II: FINE ALIGNMENT VIA MESH-BASED OPTIMIZATION

The coarse voxel placement provides pose initializations $\{\tilde{\pi}_i\}$. We refine each object's pose in the high-resolution mesh domain to ensure exact contact geometry and high visual fidelity. For object $i$ we solve:

$$\min_{\pi_i} \; E_{\text{ICP}}(\pi_i) \; + \; \mu \, E_{\text{coll}}(\pi_i; \{\pi_{j \neq i}\}) \; + \; \eta \, E_{\text{phys}}(\pi_i), \tag{11}$$

where

- $E_{\text{ICP}}(\pi_i)$ is an Iterative Closest Point (ICP) style term aligning $\mathcal{M}_i$ to target support/neighbor geometry (e.g., table surface or neighboring object contact patches):

$$E_{\text{ICP}}(\pi_i) = \sum_{v \in \mathcal{V}(\mathcal{M}_i)} \rho\big(\text{dist}(\pi_i(v), \mathcal{S}_{\text{target}})\big),$$

  with $\rho$ a robust penalty and $\mathcal{S}_{\text{target}}$ a set of scene surfaces/neighbor meshes.
- $E_{\text{coll}}(\pi_i; \{\pi_{j \neq i}\})$ penalizes mesh-level penetration with other objects (e.g., summed vertex penetration depths or triangle-triangle distances).
- $E_{\text{phys}}(\pi_i)$ enforces physical plausibility constraints such as support stability (center of mass over support polygon), uprightness or semantic orientation priors.

We optimize Eq. 11 using local nonlinear solvers (e.g., Gauss–Newton or LBFGS) combined with projective ICP steps. Importantly, $E_{\text{coll}}$ is computed on a narrow band of vertices near contacts to keep the optimization efficient.

## 4 EXPERIMENT

In this section, we conduct both qualitative and quantitative experiments to validate the effectiveness of our proposed approach. We compare our method against several representative baselines and evaluate performance with widely used metrics.

### 4.1 SETUP

**Baselines** We consider three recent 3D scene generation methods as baselines: **Gen3DSR**(Ardelean et al. (2025)), **MIDI-3D**(Huang et al. (2025)), and **PartPacker**(Tang et al. (2025)). These methods represent state-of-the-art techniques in 3D scene generation and reconstruction, providing a solid foundation for comparison.

**Metrics** To assess generation quality, we adopt three widely used metrics following remarkable 3D scene generation methods(Yao et al. (2025)): (i) CLIP score, which measures semantic alignment between the input image and the generated scene; (ii) VQ score, which reflects the visual quality and mesh fidelity; and (iii) GPT-4 metric, which prompts us to rate semantic fidelity, object arrangement plausibility, and visual realism.

**Implementation details** For evaluation, we constructed a benchmark set of 60 single-view images covering diverse domains, including indoor living spaces, outdoor urban streets, and synthetic object-centric scenes. Each method is applied to generate corresponding 3D scenes. To ensure a fair comparison, we standardize several settings: (i) all methods are run with the same number of denoising steps (50 steps); (ii) meshes are voxelized and extracted at a fixed resolution of $384^3$ to balance quality and efficiency; (iii) all meshes are simplified to 50k faces using decimation. In addition, we normalize scale and orientation by aligning generated meshes to a canonical unit cube and grounding them to the floor plane.

## 4.2 QUANTITATIVE EVALUATION

| Method | CLIP↑ | VQ↑ | GPT-4↓ | Runtime↓ |
|--------|-------|-----|--------|----------|
| MIDI-3D | 0.642 | 1.85 | 1.80 | 50s |
| Gen3DSR | 0.573 | 2.03 | 2.25 | 6min |
| PartPacker | 0.671 | 2.14 | 1.315 | 8s |
| Ours | 0.684 | 2.301 | 1.025 | 5s |

Table 1: Quantitative comparisons on CLIP score, VQ score, GPT-4 score, and runtime.

Results are shown in Table 1, and it is shown that our method consistently outperforms the baselines across all three metrics. In particular, a decrease in GPT-4 score demonstrates that our approach optimizes layout and object relationships to produce more accurate and plausible 3D scenes, while the gains in CLIP and VQ scores highlight stronger semantic consistency and higher visual fidelity.

## 4.3 QUALITATIVE EVALUATION

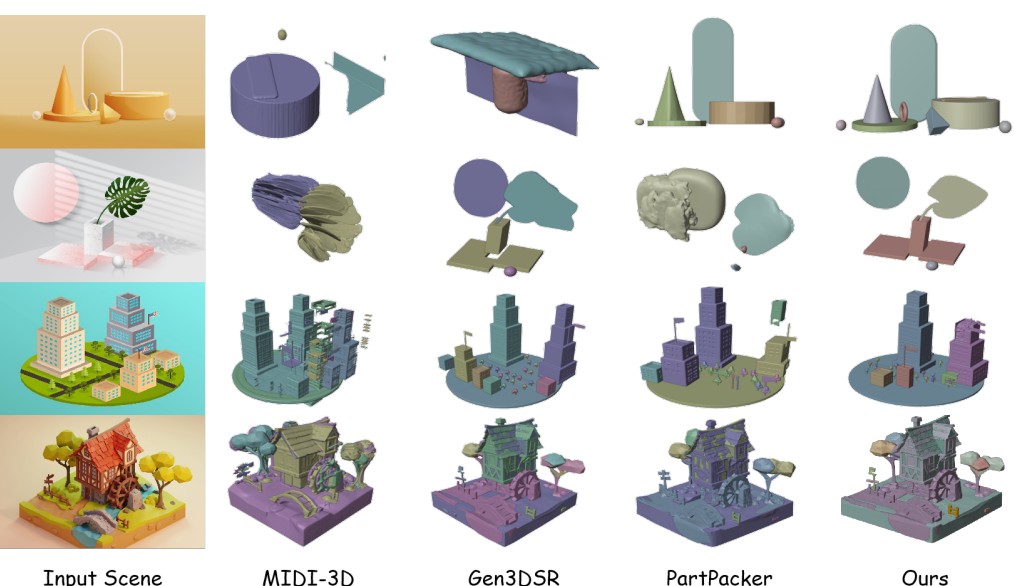

Input Scene     MIDI-3D     Gen3DSR     PartPacker     Ours

Figure 4: Qualitative comparisons on 3D Scene Generation.

Beyond quantitative metrics, we conduct extensive qualitative evaluations to highlight the strengths of our approach. Figure 4 compares scenes generated by our method with those from state-of-the-art baselines (Ardelean et al. (2025); Huang et al. (2025); Tang et al. (2025)). Our framework consistently produces higher-quality results across a variety of challenging scenarios.

First, the objects in our generated scenes exhibit more complete geometry and higher-fidelity textures, avoiding common artifacts such as over-smoothed surfaces or texture distortions. Second, our preprocessing module ensures robust extraction of objects even when the background color closely matches the object color, a case where existing methods often fail by either eroding object boundaries or introducing background leakage. Third, our appearance consistency optimization leads to improved multi-view coherence, reducing edge jitter and color mismatch across different viewpoints. Finally, the proposed voxel–mesh hybrid localization guarantees physically plausible object layouts, effectively suppressing collisions, floating artifacts, and unrealistic placements.

Together, these advantages yield scenes that are not only visually more realistic, but also structurally more coherent and semantically faithful to the input conditions. This is further corroborated by our

user study, where human raters consistently preferred our results over competing baselines in terms of geometry accuracy, appearance realism, and physical plausibility.

### 4.3.1 ABLATION STUDY

| Object Region Extraction | Color Spill Supression | Appearance Optimization | CLIP↑ | VQ↑ | GPT-4↓ |
|---|---|---|---|---|---|
| × | ✓ | ✓ | 53.14 | 1.733 | 2.75 |
| ✓ | × | ✓ | 57.26 | 1.847 | 2.375 |
| ✓ | ✓ | × | 62.67 | 2.046 | 1.875 |
| ✓ | ✓ | ✓ | 68.43 | 2.301 | 1.025 |

Table 2: Ablation study of preprocessing and appearance optimization. "✓" denotes the module is used and"x" means not used

To understand the role of Sec. 3.2, we perform a qualitative ablation focusing on the visual fidelity of the generated scenes. Table 2 shows that with each innovation removed, performance decreased according to different evaluation metrics, representing the effectiveness and validity of our module.

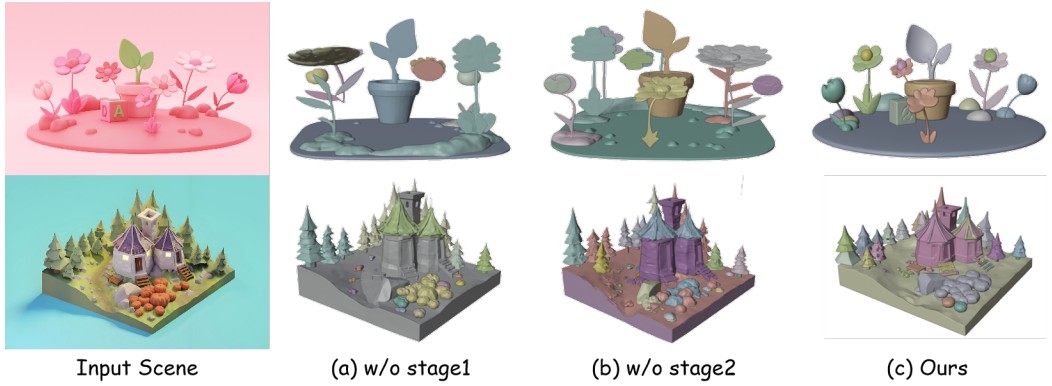

Input Scene  (a) w/o stage1  (b) w/o stage2  (c) Ours

Figure 5: Ablation on Two-Stage Optimization. From left to right: Input Scene, **w/o Stage 1**, **w/o Stage 2**, and the **full pipeline (Stage 1+Stage2)**.

According to ablation study on voxel-mesh localization shown in Figure 5, Stage 1 performs **global coarse alignment**, establishing the correct relative scale, orientation, and inter-object distances. This early adjustment prevents large-scale inconsistencies that would otherwise propagate to later refinements. Stage 2 focuses on **fine alignment via mesh-based optimization**, including sub-mesh deformation, texture completion, and collision-aware placement.

Removing Stage 1 causes the optimization in Stage 2 to struggle with global drift, while removing Stage 2 leaves subtle penetrations and floating artifacts unresolved. The complete pipeline, therefore, benefits from the complementary strengths of both stages.

## 5 CONCLUSION

We presented Geo-Refine, a single-image 3D scene generator that fuses geometry–appearance pre-processing with a voxel–mesh localization pipeline. By pairing projection-aware placement with physical priors and fine mesh alignment, the method delivers strong semantic consistency, structural coherence, and visual fidelity across varied scenarios. Looking ahead, we plan to extend this framework to dynamic scenes and interactive editing, moving closer to real-time, physically grounded 3D world modeling.

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

# A APPENDIX

## A.1 THE USE OF LLM

We employed large language model (ChatGPT, GPT-5) **solely for language refinement** after completing the research and drafting the manuscript. The model assisted with grammar correction, clarity improvements, and minor stylistic edits. **No text, data, or ideas were generated beyond these surface-level edits**, and all substantive content—conceptualization, methodology, analysis, and conclusions—was created entirely by the authors. The authors take full responsibility for the final manuscript.

## A.2 DETAILS OF GEOMETRIC AND APPEARANCE PREPROCESSING

### A.2.1 HIGH-PRECISION OBJECT REGION EXTRACTION

Existing segmentation method (Yao et al. (2024)) achieve pixel-level accuracy but remain vulnerable to boundary erosion in cluttered scenes and lack explicit scene-level priors, leading to ambiguous object masks. To overcome this, we formulate object segmentation as a prior-constrained optimization problem:

$$\hat{M}_j = \arg\min_{M_j} \ \mathcal{L}_{\text{seg}}(M_j, I) + \lambda \, \mathcal{L}_{\text{spatial}}(M_j, \mathcal{R}), \tag{12}$$

where $M_j$ is the mask of object $j$, $\mathcal{L}_{\text{seg}}$ ensures fidelity to the image $I$, and $\mathcal{L}_{\text{spatial}}$ incorporates relational priors $\mathcal{R}$ (e.g., object–ground contact, occlusion hierarchy).

To further refine geometry, we integrate object-aware sampling that emphasizes high-curvature regions and boundary-focused refinement that iteratively sharpens mask edges. This approach preserves fine-grained object boundaries and reduces over-eroding effects, providing more faithful inputs for downstream 3D reconstruction.

### A.2.2 DIRECTIONAL COLOR SPILL SUPPRESSION

Conventional background removal and chroma-keying approaches typically treat color spill as a global correction problem, which is insufficient in real-world imagery where reflective surfaces and environmental lighting cause directional, non-uniform color bleeding. To address this, we propose a **direction-aware spill suppression mechanism**, modeling observed colors as:

$$C(x) = C_{\text{obj}}(x) + \alpha(x) \cdot S(x), \quad x \in M_i, \tag{13}$$

where $C_{\text{obj}}(x)$ is the intrinsic texture, and $S(x)$ is the spill component modulated by a coefficient

$$\alpha(x) = \sigma\left(\nabla I(x) \cdot d_{\text{light}}(x)\right), \tag{14}$$

with $\nabla I(x)$ denoting the local gradient and $d_{\text{light}}(x)$ the estimated illumination direction.

This mechanism selectively attenuates unwanted hues while preserving intrinsic textures. In practice, illumination-conditioned filtering and contrast-preserving correction are applied, yielding sharper geometry and higher-fidelity material appearance than global correction strategies.

### A.2.3 MULTI-VIEW APPEARANCE CONSISTENCY OPTIMIZATION

In multi-view scenarios, existing methods (Mildenhall et al. (2020); Liu et al. (2021)) primarily focus on geometric consistency while neglecting appearance harmonization, resulting in color shifts and edge jittering across views. To mitigate this, we introduce a **feature-level alignment and normalization scheme**. Let $F_v^i$ denote feature embeddings of object $i$ under view $v$; we minimize inter-view discrepancies via:

$$\mathcal{L}_{\text{cons}} = \sum_{(v_1, v_2)} \left\| \phi(F_{v_1}^i) - \phi(F_{v_2}^i) \right\|_2^2, \tag{15}$$

where $\phi(\cdot)$ encodes both appearance and boundary features.

We further apply a color normalization layer to enforce consistent mean and variance across views. In video inputs, temporal coherence regularization is introduced to suppress jittering and flickering. Together, these techniques significantly enhance mesh alignment accuracy and produce coherent textures in downstream 3D synthesis.

By integrating high-precision extraction(Yao et al. (2024)), directional spill suppression, and multi-view harmonization (e.g. optimization methods like (Zhao et al. (2024))), our preprocessing module produces object inputs that are both geometrically accurate and visually coherent. This forms a strong foundation for downstream voxel-mesh hybrid localization.

