# OpenReview forum: "Geo-Refine: Geometry–Appearance Synergy for Robust Single-Image 3D Scene Generation"
_ICLR.cc/2026/Conference — ICLR 2026 Conference Withdrawn Submission_

### Official Review · Reviewer_M2Vk · 2025-10-31

**Soundness:** 2
**Presentation:** 3
**Contribution:** 3
**Rating:** 4
**Confidence:** 2

**Summary:**

An image -> 3D paper that works with a novel part based architecture and multi-stage generation that helps with feasibility of the generations to ensure a lack of collisions and good geometry that fits together nicely with good contact points and everything.

**Strengths:**

I like the concept and direction of the work overall, with good qualitative results that give me some idea of the quality of the method, I think the problem and method are well motivated which I enjoy.

**Weaknesses:**

My main concern is the data used and comparisons to existing works are lacking and hindering the clarity and significance of the work. It's not clear to me what this dataset looks like or how the size of it compares to object level datasets that are common like objaverse. I'd also like more comparisons with existing works in 3D generation. Lots of object level 3D generation techniques exist, and I know that most of those like TRELLIS for instance aren't particularly "part aware" and don't really explicitly account for feasibility, but I still feel like I need to understand how this fits into the existing landscape of object level generative 3D. The metrics don't lend themselves to me understanding this work in the broader landscape of existing work.

**Questions:**

Can you help me understand 1) the dataset in question 2) the situation with baselines or how this work can be compared to other existing works to inform me of why I would use this method compared to existing object level 3D generative methods that don't take into account feasibility, or some more quantitative results that might be able to demonstrate to me that this line of work if continued may yield better results than what we have in existence right now.

---

### Official Review · Reviewer_88ws · 2025-10-31

**Soundness:** 3
**Presentation:** 3
**Contribution:** 3
**Rating:** 4
**Confidence:** 4

**Summary:**

The paper proposes Geo-Refine, an end-to-end single-image to 3D scene generation framework that couples a geometry–appearance preprocessing module with a two-stage voxel–mesh localization pipeline.

**Strengths:**

1. Well-motivated pipeline: The paper clearly identifies two real pain points in single-image to 3D scene generation: (i) reliance on external segmentation priors, and (ii) sequential per-mesh processing is inefficient, with inference cost scaling linearly with the number of meshes. Geo-Refine attacks exactly these two.
2. Experiments: Runtime is a tangible practical benefit, especially if this needs to be used in interactive or dataset-scale generation. Ablation experiments supports the authors' claim well.
3. The paper is well-written and easy to follow.

**Weaknesses:**

1. Evaluation scale is small and a bit private: the benchmark is 60 images, “diverse indoor and outdoor” but not clearly tied to a public, reproducible dataset. This makes it hard to judge how robust the method is, or to compare future work to it. This is the biggest concern.
2. The qualitative result is not plausible: In the 3rd case of Fig.4, it seems that all of the methods including the proposed method fail to generate reasonable geometry.
3. Engineering-heavy contribution: a lot of the gain seems to come from carefully designed, non-learned steps (mask optimization with priors, directional color-spill model, etc.). That’s fine for a system paper, but for ICLR the central learning idea is a bit weak.

**Questions:**

My concerns are listed in weaknesses part.

---

### Official Review · Reviewer_6yPD · 2025-11-01

**Soundness:** 2
**Presentation:** 2
**Contribution:** 2
**Rating:** 4
**Confidence:** 3

**Summary:**

Geo-Refine proposes an end-to-end framework for generating 3D scenes from a single RGB image with compositional, multi-object outputs. The method couples geometry–appearance preprocessing with a dual-stream transformer and a two-stage voxel–mesh localization scheme. The model is evaluated on a benchmark of 60 single-view examples, showing improved visual quality and semantic consistency compared to recent single-image 3D generation baselines.

**Strengths:**

The paper addresses an important and timely task in single-image 3D scene reconstruction. The proposed model demonstrates visually promising results and shows clear qualitative improvements over existing baselines.

**Weaknesses:**

I found the the paper a bit hard to follow:

- The training setup is not described at all. It is unclear which datasets were used, what supervision signals are available, and how the flow and placement modules are optimized.

- The 3D decoder is only mentioned but never explained — there is no description of its representation (e.g., occupancy, implicit field, voxel, or mesh) or how the final meshes are generated from latent features.

- The paper claims to produce collision-free and physically plausible results, but these claims are not supported by quantitative or visual evaluation.

- The evaluation section is very limited, with many proposed contribution not ablate properly, hard to understand their contribution.

**Questions:**

See Weaknesses

---

### Official Review · Reviewer_BoGC · 2025-11-02

**Soundness:** 2
**Presentation:** 1
**Contribution:** 2
**Rating:** 2
**Confidence:** 4

**Summary:**

This paper proposes Geo-Refine, a framework for generating multi-object 3D scenes from a single RGB image through a geometry–appearance synergy pipeline. It introduces three components: a geometry–appearance preprocessing stage; a dual-stream latent encoder based on DINOv2 and a transformer that produces geometry and relation tokens; and a two-stage voxel–mesh localization. Evaluated with CLIP, VQ, and GPT-4 metrics, Geo-Refine is reported to improve visual realism, semantic consistency, and efficiency over previous single-image 3D scene generation methods.

**Strengths:**

The problem scope and proposed direction are significant. The paper tackles an important and timely problem—multi-object 3D scene generation from a single RGB image—and presents an ambitious attempt to integrate geometric reasoning, appearance consistency, and physical plausibility within a unified framework. Its originality lies in formulating a geometry–appearance synergy and introducing a two-stage voxel–mesh refinement process that explicitly targets structural and physical consistency, which is rarely emphasized in prior single-view 3D reconstruction work. The idea of combining high-level visual features (via DINOv2) with energy-based spatial optimization is conceptually appealing and has potential to inspire future research bridging 2D visual understanding and 3D physical reasoning.

**Weaknesses:**

The paper is conceptually ambitious but its presentation severely lacks methodological clarity. Every stage is described in abstract, high-level terms without clear input–output definitions or concrete data flow, making the method extremely difficult to understand or reproduce.
The paper never clearly defines what data are passed between stages. It is impossible to trace how an RGB image becomes 3D meshes. For example, the inputs and outputs of each stage (preprocessing, DINOv2 encoding, transformer, voxel–mesh localization, decoder) are not explicitly described or shown in a figure. There is no clear description of what intermediate representations look like — e.g., features, object–ground contact, occlusion hierarchy, etc. The geometry–appearance preprocessing module supposedly feeds into DINOv2, but the interface between them (image crops, binary masks, or feature tensors) is undefined. The mesh decoder’s input and output formats (e.g., SDF, occupancy, vertex list, or mesh templates) are not stated.


The evaluation section is non-standard and under-specified. There is no quantitative measure of geometry accuracy (e.g., Chamfer, IoU, or collision metrics), even though geometry quality is the paper’s main claim. The metrics (CLIP, VQ, and GPT-4) are borrowed from image-based works but not adapted to 3D: the paper has not explained how scenes are rendered, which views are sampled, or how text prompts are formed for CLIP/GPT-4. VQ seems to be a subjective user study from the referenced paper, but the protocol is unspecified. The dataset details are also unclear — whether they are real or synthetic, how they were collected or rendered, whether there is ground truth 3D information, etc. The ablation study is highly sparse, with a mixture of quantitative and qualitative comparison, instead of unified quantitative metrics across all studies. What datasets are used for training also seem to be missing.

**Questions:**

Please refer to weaknesses for questions raised due to unclarity of the paper's original presentation

---

### Note · Authors · 2025-11-13

I have read and agree with the venue's withdrawal policy on behalf of myself and my co-authors.